# Real-time chirality transfer monitoring from statistically random to discrete homochiral nanotubes

Shixin Fa [1,2,6], Tan-hao Shi[1,6], Suzu Akama[1], Keisuke Adachi[1], Keisuke Wada[1], Seigo Tanaka[1], Naoki Oyama[1], Kenichi Kato [1], Shunsuke Ohtani[1], Yuuya Nagata [3], Shigehisa Akine [4,5] & Tomoki Ogoshi [1,4] ✉

Real time monitoring of chirality transfer processes is necessary to better understand their kinetic properties. Herein, we monitor an ideal chirality transfer process from a statistically random distribution to a diastereomerically pure assembly in real time. The chirality transfer is based on discrete trimeric tubular assemblies of planar chiral pillar[5]arenes, achieving the construction of diastereomerically pure trimers of pillar[5]arenes through synergistic effect of ion pairing between a racemic rim-differentiated pillar[5]arene pentaacid bearing five benzoic acids on one rim and five alkyl chains on the other, and an optically resolved pillar[5]arene decaamine bearing ten amines. When the decaamine is mixed with the pentaacid, the decaamine is sandwiched by two pentaacids through ten ion pairs, initially producing a statistically random mixture of a homochiral trimer and two heterochiral trimers. The heterochiral trimers gradually dissociate and reassemble into the homochiral trimers after unit flipping of the pentaacid, leading to chirality transfer from the decaamine and producing diastereomerically pure trimers.

Chirality transfer, in which chiral information is transmitted from one molecule (or part of it) to others (and/or from one size scale to another) is of crucial importance in many physical and biological processes as well as in chemistry and materials science[1–4]. Chirality transfer has been achieved in a variety of artificial systems, such as polymeric systems[5–7], supramolecular polymers[8–14], and discrete assemblies[15–19]. For instance, Yashima et al. prepared right- and left-handed helical polymers based on chirality transfer from chiral amines or alcohols to the polymeric backbones through hydrogen bonding interactions between acid groups on the polymer side chains and the chiral inducers[20]. This chirality transfer was performed by forming single-stranded spiral polymer chains. Furthermore, Meijer et al. developed chirality transfer systems with *P*- and *M*-helical structures

based on π-π stacking assemblies, where aromatic molecules with chiral substituents were used as chiral inducers[21,22]. In these systems, the chirality was transferred to achiral building blocks with similar molecular structures via π-π stacking of aromatic rings, producing helical assemblies. In these cases, the chiral information was amplified by chirality transfer from the chiral inducers to the achiral acceptors, creating chiral supramolecular structures through the subsequent assembly. Moreover, these groups also conducted real-time monitoring of the chirality transfer process in these systems to better understand their kinetic properties (Fig. 1a)[23,24]. Due to the complexity of polymeric systems, there are countless possible intermediate assemblies during the chirality transfer process. Therefore, chirality transfer processes as observed in real time typically reflect the averaged

[1]Department of Synthetic Chemistry and Biological Chemistry, Graduate School of Engineering, Kyoto University, Katsura, Nishikyo-ku, Kyoto 615-8510, Japan. [2]School of Chemistry and Chemical Engineering, Northwestern Polytechnical University, Xi'an, Shaanxi 710072, P.R. China. [3]WPI Institute for Chemical Reaction Design and Discovery (WPI-ICReDD), Hokkaido University, Kita 21 Nishi 10, Kita-ku, Sapporo 060-0810, Japan. [4]WPI Nano Life Science Institute (WPI-NanoLSI), Kanazawa University, Kakuma-machi, Kanazawa 920-1192, Japan. [5]Graduate School of Natural Science and Technology, Kanazawa University, Kakuma-machi, Kanazawa 920-1192, Japan. [6]These authors contributed equally: Shixin Fa, Tan-hao Shi. ✉e-mail: ogoshi@sbchem.kyoto-u.ac.jp

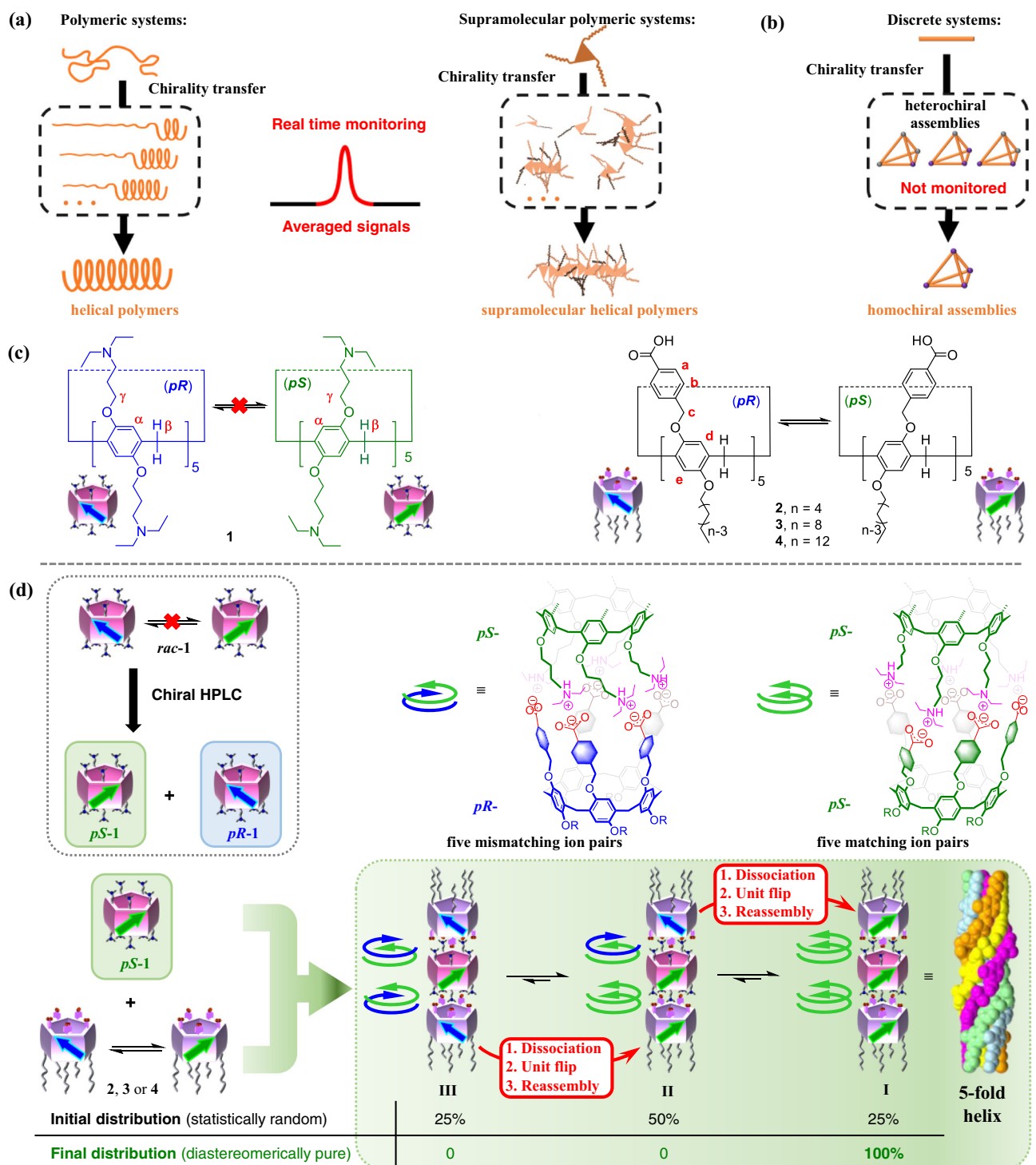

**Fig. 1 | Real-time monitoring of reported chirality transfer systems and that in the present study. a** Real-time monitoring of chirality transfer in polymeric and supramolecular polymeric systems, which typically reflect the averaged properties of each intermediate assembly in the system. **b** Discrete chirality transfer systems, which are difficult to monitor in real-time. **c** Chemical structures and planar chirality of pillar[5]arenes **1**–**4**. **d** The illustration of the process of statistically random to diastereomerically pure discrete trimeric nanotube through chirality transfer by matching ten ion pairs between *pS*-**1** and **2** (or **3**, **4**).

properties of each intermediate assembly in the system, making monitoring chirality transfer accurately in real-time challenging.

Compared with polymeric and supramolecular polymeric systems involving chirality transfer, discrete chirality transfer systems are much easier to monitor and analyze because of their simpler structures[15–19]. For example, enantiopure metallocages and organic cages can be synthesized using chirality transfer from chiral guest molecules or chiral auxiliary ligands[25–30]. However, these initial studies

mainly focused on the chirality transfer in the final stage (equilibrium state), because controlling the chirality transfer rate and trapping the heterochiral assemblies were difficult (Fig. 1b). In several studies, a small amount of chiral ligand was found to induce the immediate formation of homochiral cages[31–33]. Directional interactions, such as covalent bonds, imine bonds, hydrogen bonds, and metal coordination, were mainly used to create these discrete assemblies. The directional interactions of several ligands or moieties must cooperate

to construct a detectable assembly, which makes the heterochiral structures in metastable states difficult to observe. By weakening the interactions during the assembly process, thereby reducing its directivity, some metastable heterochiral assemblies may be observed, but real-time monitoring of such transitions remains difficult[34]. On the other hand, both homochiral assemblies and their heterochiral isomers can be observed at the beginning of mixing in several self-assembled systems, but not by a chirality transfer strategy[35]. Accordingly, to the best of our knowledge, monitoring a chirality transfer process from a statistically random assembly including heterochiral and homochiral units to a diastereomerically pure and thermodynamically stable homochiral assembly in a non-averaged discrete assembled system, i.e., a general and ideal chirality transfer process, remains challenging.

In our work investigating chirality transfer systems based on discrete tubular assemblies of planar chiral pillar[5]arenes, we are able to monitor the chirality transfer from a statistically random distribution to a diastereomerically pure assembly in real time. By rational molecular design of the pillar[5]arenes, the initial states of the statistically dependent homo- and heterochiral supramolecular assembly mixture was trapped by using strong and non-directional ionic interactions, and the chirality transfer from the heterochiral to homochiral state proceeded as a result of the directionality of substituents on the planar chiral pillar[5]arenes.

Pillar[5]arenes are pillar-shaped macrocyclic host molecules[36–41]. They have two stable planar chiral conformers owing to directional substitutions on their phenyl units (Fig. 1c)[42]. In general, the two conformers interconvert because of unit flipping[43–45]. However, introducing bulky groups to the rims can slow or even stop this flipping, producing two atropisomers[43,46,47].

Recently, we have prepared peraminopillar[5]arene 1, which bears ten 3-(diethylamino)propoxy groups on its rims, and rim-differentiated acidic pillar[5]arenes 2, 3, and 4, which have five benzoic acids on one rim and five linear alkyl chains on the other (Fig. 1c)[48,49]. By taking advantage of the strong ionic interactions between the amino groups on 1 and the acid groups on 2, 3, and 4, mixtures of 1 and 2 (or 3, 4) in a 1:2 molar ratio gave rise to trimeric nanotubular assemblies. However, details of this assembly process, including the chiral information for each component, proved difficult to discern. Subsequently, we tried to use acidic pillar[5]arenes containing five stereogenic carbons to achieve the construction of chiral nanotubes by regulating the diastereomeric excess of the acidic pillar[5]arenes using guest molecules, but the structural details of the chiral assemblies were still not clear[50].

Here, we show the chirality transfer of the trimeric nanotubular assemblies and the construction of diastereomerically pure trimeric nanotubes obtained as thermodynamically stable products when chiral peraminopillar[5]arene 1 is used as a chiral building block (Fig. 1d). More gratifyingly, we successfully control the rate of the chirality transfer process by changing the length of the alkyl chains on the acidic pillar[5]arene, enabling real-time monitoring the process from statistically random to diastereomerically pure assembly. When pS-1 was mixed with racemic 4, which can undergo unit flipping, the metastable heterochiral diastereomers pR-pS-pS and pR-pS-pR (i.e., trimers II and III) were initially trapped along with the thermodynamically stable homochiral pS-pS-pS (i.e., trimer I), according to a statistically random distribution, because ionic interactions are generally strong and non-directional. However, the five ion pairs of the directional rims between pS-acidic and pS-basic pillar[5]arenes are principally more stable than those between pR-acidic and pS-basic pillar[5]arenes because of the planar chirality matching. Therefore, the mismatching ion pairs between pR-4 and pS-1 dissociate gradually, the units of pR-4 flip from the pR- to the pS-form[51], and the resulting pS-acidic pillar[5]arenes reassemble with pS-1 by matching ion pairs. Finally, only the thermodynamically stable homochiral trimer pS-pS-pS

(i.e., I) is observed, producing a tubular structure with a fivefold helix. The chirality transfer from trimers II and III to I can be clearly monitored by ¹H NMR measurements. In this work, we report the real-time monitoring of chirality transfer from a statistically distributed mixture to a disateromerically pure nanotubular assembly.

## Results
### Optical resolution of 1
In the ¹H NMR spectrum of peraminopillar[5]arene 1, the $H_\gamma$ protons on the two rims in proximity to the pillar[5]arene core show a clear AB-quartet split, which indicates that the two protons on the same ethylene group are diastereotopic and evidences the inhibition of free rotation of the pillar[5]arene units at room temperature (Supplementary Fig. 1a)[43]. When mixed with 10 equiv. of D-mandelic acid (D-MA), a chiral solvating agent used to discriminate the signals of optically active analytes in NMR spectroscopy[52], the $H_\alpha$ signal on the core of 1 is split into two singlet peaks with equal integration values, implying the formation of supramolecular diastereomers between D-MA and pS-/pR-1 via ionic interaction (Supplementary Fig. 1b). This observation further reveals that the two enantiomers (i.e., pS- and pR-1) are stable and do not readily interconvert at room temperature owing to the bulkiness of the ten 3-(diethylamino)propoxy groups on the rim. Optical resolution of 1 was realized by chiral HPLC (Supplementary Fig. 2). The two fractions were found to be highly pure, as evidenced by the HPLC traces of the two reinjected fractions (Supplementary Fig. 3) and the ¹H NMR spectra of the two fractions in the presence of D-MA (Supplementary Figs. 4 and 5). The CD spectra for the two fractions are mirror images, indicating their enantiomeric relationship (Supplementary Fig. 6). Based on the Cotton effect at ca. 310 nm, which is ascribed to the π-π* transition in the pillar[5]arene backbones[53], the first and second fractions were assigned as pR-1 and pS-1, respectively[54]. The enantiomeric 1 is stable at room temperature as the Gibbs energy of activation at 25 °C ($\Delta G^\ddagger_{25°C}$) was determined to be 101 kJ/mol by Eyring plot at 40–55 °C. Moreover, the CD intensity of enantiomeric 1 does not significantly decrease over four weeks at room temperature (see Supplementary Section 2.2 for details). However, too high a temperature will rapidly racemize enantiomeric 1, as the CD signal of enantiomeric 1 at 100 °C in tetrachloroethane completely disappeared within 3 h, and no signal was observed even when it cooled back to room temperature (Supplementary Fig. 9).

### Chirality transfer from enantiomeric 1 to 2 to form homochiral trimers
Mixing of enantiomeric 1 with rim-differentiated acidic pillar[5]arene 2 in a 1:2 molar ratio gives rise to the formation of trimeric assemblies, as observed for a previously reported racemic mixture (Supplementary Figs. 10 and 11)[48]. The intensity of the UV absorption for the pillar[5]arene π-π* transition is approximately tripled because the concentration of the pillar[5]arene core increases by a factor of two (Fig. 2a). However, the corresponding CD intensity increase is approximately fivefold. This result strongly implies chirality transfer and amplification in the mixture during trimer formation. Thus, 2 becomes non-racemic and exhibits an enantiomeric excess (ee) through chirality transfer from enantiomeric 1, which makes the formation of the homochiral trimers (i.e., I and its enantiomeric trimer) more favorable.

Nevertheless, the fivefold increment in CD intensity (instead of threefold) can only be achieved with contributions by other factors. We propose that the additional increase in CD intensity is due to structural fixing caused by trimerization (Fig. 2b). Although unit flipping of 1 is inhibited at room temperature by the bulky substituents on both rims, the non-rigid methylene bridges of 1 allow the units to swing while maintaining the planar chirality of the molecule. This swing effect means the pillar[5]arene molecules may not always exist in a perfectly pillar-shaped configuration in the solution, and thus their apparent CD signals may not be maximized, which is quite different

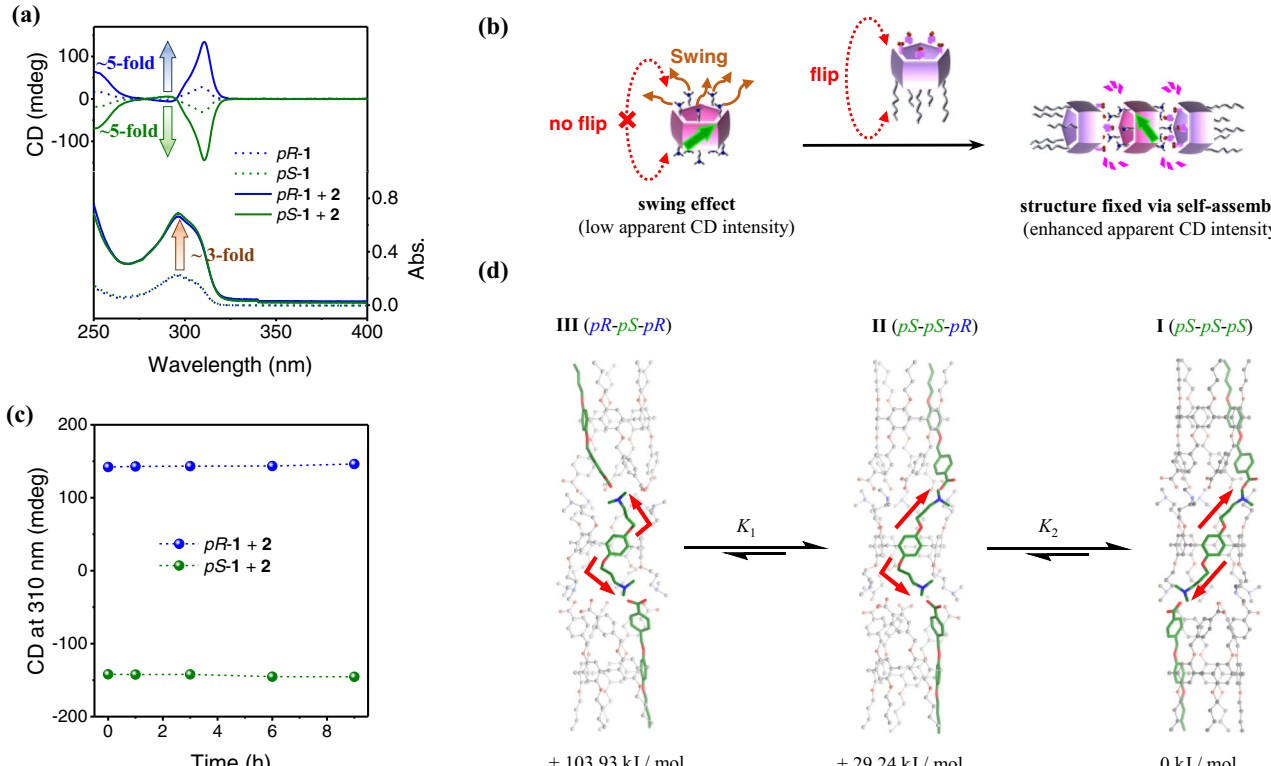

**Fig. 2 | Spectral results and proposed chirality transfer from enantiomeric 1 to 2 by trimerization. a** UV–Vis and CD spectra (chloroform, 25 °C) of pR-**1**, pS-**1**, and mixtures of **2** with pR-**1** or pS-**1**. The concentrations of **1** and **2** were 0.05 and 0.1 mM, respectively. **b** Schematic of the swing effect of pS-**1** on CD intensity and structure fixing via trimerization to suppress the swing effect. **c** CD intensity at 310 nm of mixtures of pR-**1** and pS-**1** with **2** upon heating at 50 °C. The concentrations of **1** and **2** were 0.05 and 0.1 mM, respectively. **d** Optimized structures and relative energies of the three possible trimers of the mixture of **2** with pS-**1** in chloroform (ORCA 4.2.0 with the semiempirical GFN2-xTB method and the SMD solvation model)[56–58].

from most chiral molecules. Accordingly, we verified this swing effect experimentally using host-guest complexation, which suppresses the unit swing of pillar[5]arenes. Upon the addition of 1,4-dibromobutane, a known guest for the cavity of pillar[5]arenes ($K > 10^3 M^{-1}$)[55], the CD intensity of enantiomeric **1** increases by up to 50% (Supplementary Fig. 13).

Similarly, trimerization of **1** with **2** also fixes the structure of the pillar[5]arenes through multiple ionic interactions. Therefore, an additional CD increase of the system (Fig. 2a) beyond the effect of chirality transfer is observed (see Supplementary Section 3.2 for details of discussion on the swing effect).

The amplified CD intensity barely changes upon heating at 50 °C for 9 h (Fig. 2c and Supplementary Fig. 14), which indicates that the chirality transfer process reaches equilibrium immediately upon mixing. To obtain distribution information on the trimers in the system, we performed computational optimization of the three trimers centered on pS-**1** (i.e., **I**, **II**, and **III** in Fig. 2d)[56–58]. Based on theoretical calculations, the homochiral trimer **I** is energetically more stable than the other two heterochiral trimers by 29.24 and 103.93 kJ/mol (see Supplementary Section 3.4 for computational details, and Supplementary Data 1 contains the optimized structures in xyz format). It is apparent that the effect of planar chirality in pillar[5]arenes lead the ten matching ion pairs of the same directional rims between the pS-acidic and pS-basic pillar[5]arenes in trimer **I** to gain more stabilization energy than those between the pR-acidic and pS-basic pillar[5]arenes in heterochiral trimers **II** and **III**. According to the Maxwell–Boltzmann distribution, **I** makes up the overwhelming proportion of the system. At room temperature, the equilibrium constants between trimer **III** and **II**, and between **II** and **I** were determined to be $1.4 \times 10^{13}$ and $1.3 \times 10^5$, respectively. Therefore, almost only homochiral trimer **I**, the thermodynamically stable species, exists in the 1:2 mixture of

enantiomeric **1** and **2**. Thus, the presence of only one set of resonance signals for the mixture of pS-**1** and **2**, which are identical to that for the mixture of rac-**1** with **2** (Supplementary Fig. 10), indicates the formation of a highly symmetrical trimerized structure (i.e., **I** in the former case, and **I** and its enantiomeric trimer in the latter).

This fast and efficient chirality transfer was not expected. Considering that ionic interactions are generally strong and non-directional, the stabilization produced by the ten ion pairs during trimerization promotes the survival of the metastable heterochiral trimers. When rim-differentiated acidic pillar[5]arene **4** with longer alkyl chains on the rim was used, the process of chirality transfer became slow, and thus we more clearly monitored it.

## Chirality transfer from enantiomeric 1 to 4 to form homochiral trimers

Unlike the case for the mixture of enantiomeric **1** and **2**, mixing of enantiomeric **1** with **4** in a 1:2 molar ratio leads to an overall CD increase of less than twofold, despite the trimerization of **1** and **4** (Fig. 3a). This observation suggests that the chiral information of **1** does not transmit to **4** as efficiently as to **2**, which is caused by the increased length of the alkyl chains on the rim of the acidic pillar[5]arene. Furthermore, this phenomenon also implies that species other than the homochiral trimer **I** may be kinetically trapped.

In the ESI-MS spectrum of the mixture of **4** and pS-**1**, signals corresponding to the ionized trimers are clearly observed (Fig. 3b and Supplementary Fig. 16). The $^1$H NMR spectrum of the same mixture shows many resonances in addition to those in the spectrum of the mixture of rac-**1** with **4** (Fig. 3c, d and Supplementary Fig. 17), suggesting a more complicated system than that for homochiral trimer **I** (orange signals in Fig. 3c). These new signals cannot be separated from those of **I** in diffusion ordered spectroscopy (DOSY). The diffusion

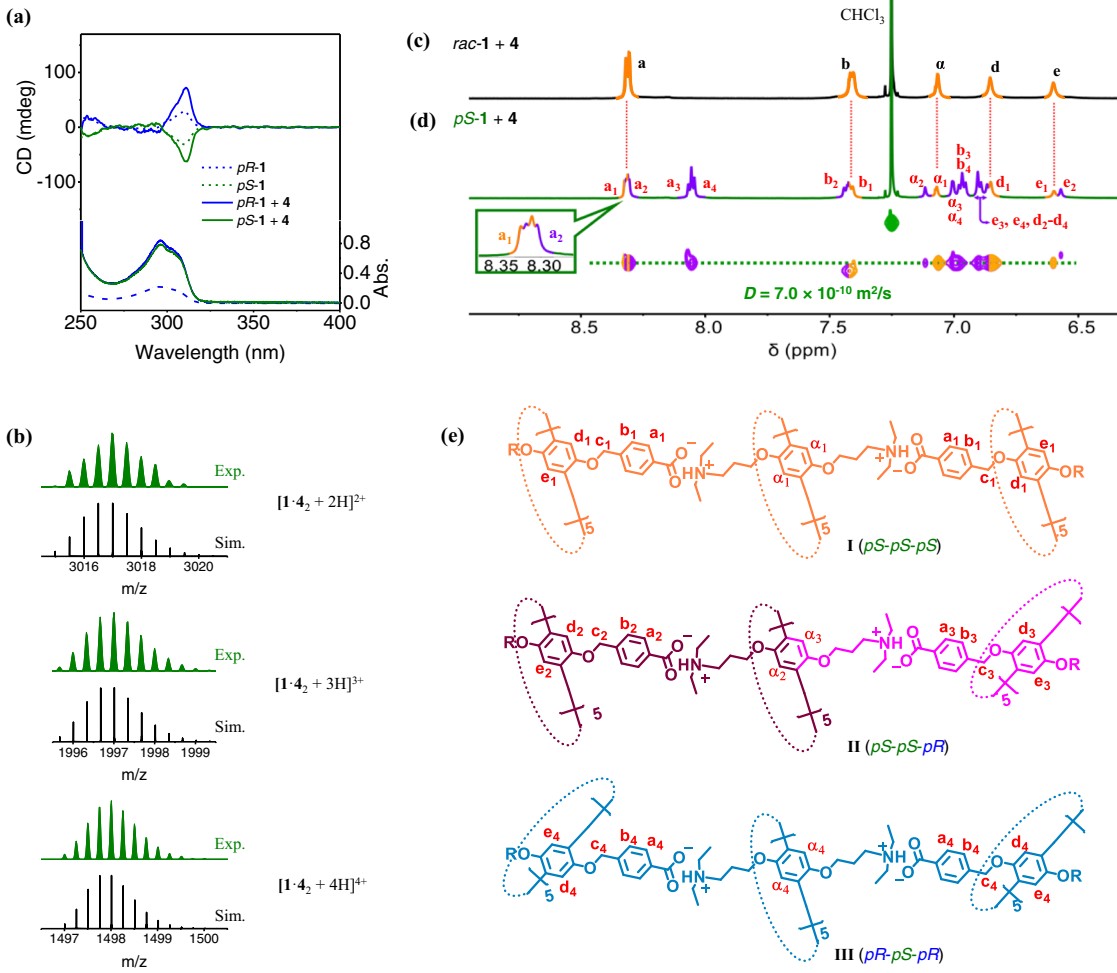

**Fig. 3 | Spectral results for a mixture of enantiomeric 1 and 4. a** UV–Vis and CD spectra (chloroform, 25 °C) of *pR*-**1**, *pS*-**1**, and mixtures of **4** with *pR*-**1** or *pS*-**1**. The concentrations of **1** and **4** were 0.05 and 0.1 mM, respectively. **b** Simulated (Sim.) and experimental (Exp.) high-resolution ESI-MS spectra of a mixture of **4** with *pS*-**1**. **c, d** Partial ¹H NMR spectra (600 MHz, CDCl₃, 25 °C) for as-prepared mixtures of **c** *rac*-**1** and **4**, and **d** *pS*-**1** and **4**. The DOSY NMR spectrum (500 MHz, CDCl₃, 25 °C) of the as-prepared mixture of *pS*-**1** and **4** is also shown in **d**. For all ¹H NMR spectra, the concentrations of **1** and **4** were 0.5 and 1.0 mM, respectively. **e** Molecular structures of the three trimers in the mixture of *pS*-**1** and **4**.

coefficient (*D*) was determined to be $7.0 \times 10^{-10}$ m²/s (Fig. 3d), which is in good agreement with the *D* value of our previously reported trimers[48], suggesting that the new species are also trimeric assemblies. We speculate that the formation of the ten salt bridges in either the homochiral or heterochiral trimers significantly decreases the free energy of the system, leading to an absence of dimers and monomers. This was further evidenced by varying the ratio of **1** and **4**. When less than 2 equiv. of **4** is added to the solution of **1**, signals corresponding to dimers and monomers are observed (Supplementary Figs. 18 and 19). By carefully comparing these spectra with that shown in Fig. 3d, it is possible to determine that only trimeric species are present in the mixture of *pS*-**1** and **4** at a 1:2 molar ratio.

The signals in the ¹H NMR spectrum were assigned convincingly based on their chemical shifts and integrations (Fig. 3d, e). In principle, the resonances of **I** and **III** present only one set of peaks each, because of the bilateral symmetry of the structure. In contrast, the structure of **II** is flanked by matching interactions and mismatching interactions, so the NMR signals for the different ends of the structure are not identical, and their chemical shifts are similar to those in **I** and **III**, respectively (Fig. 3e). Therefore, a total of four sets of NMR signals can be observed for the mixture of the three assemblies (Fig. 3d). The peak with chemical shift ~8.3 ppm clearly contains $H_a$ in **I** and another $H_a$, which can be assigned as that at the matching end of structure **II**. The

new peaks with chemical shifts ~8.1 ppm should be $H_a$ in **III** and at the mismatching end of **II**. Similarly, the two peaks ~6.6 ppm represent $H_e$ in **I** and that at the matching end of **II**. This assignment of these peaks allowed us to infer the initial concentrations of the three assemblies in the solution. As shown in Fig. 3d, e, the peak with a chemical shift of approximately 8.3 ppm contains the resonance signals of all the $H_a$ in **I** and $H_a$ at the matching end of **II**, i.e., its integral area is positively correlated with the sum of the concentrations of **II** and two times the concentration of **I**:

$$\int_{8.3\text{ppm}} \propto (2[\mathbf{I}] + [\mathbf{II}]). \tag{1}$$

In the same way,

$$\int_{8.1\text{ppm}} \propto (2[\mathbf{III}] + [\mathbf{II}]); \tag{2}$$

$$\int_{\text{He1}} / \int_{\text{He2}} = 2[\mathbf{I}]/[\mathbf{II}]. \tag{3}$$

Accordingly, the initial distribution of the three trimers was determined to be 25% for **I**, 50% for **II**, and 25% for **III**, which is in accord

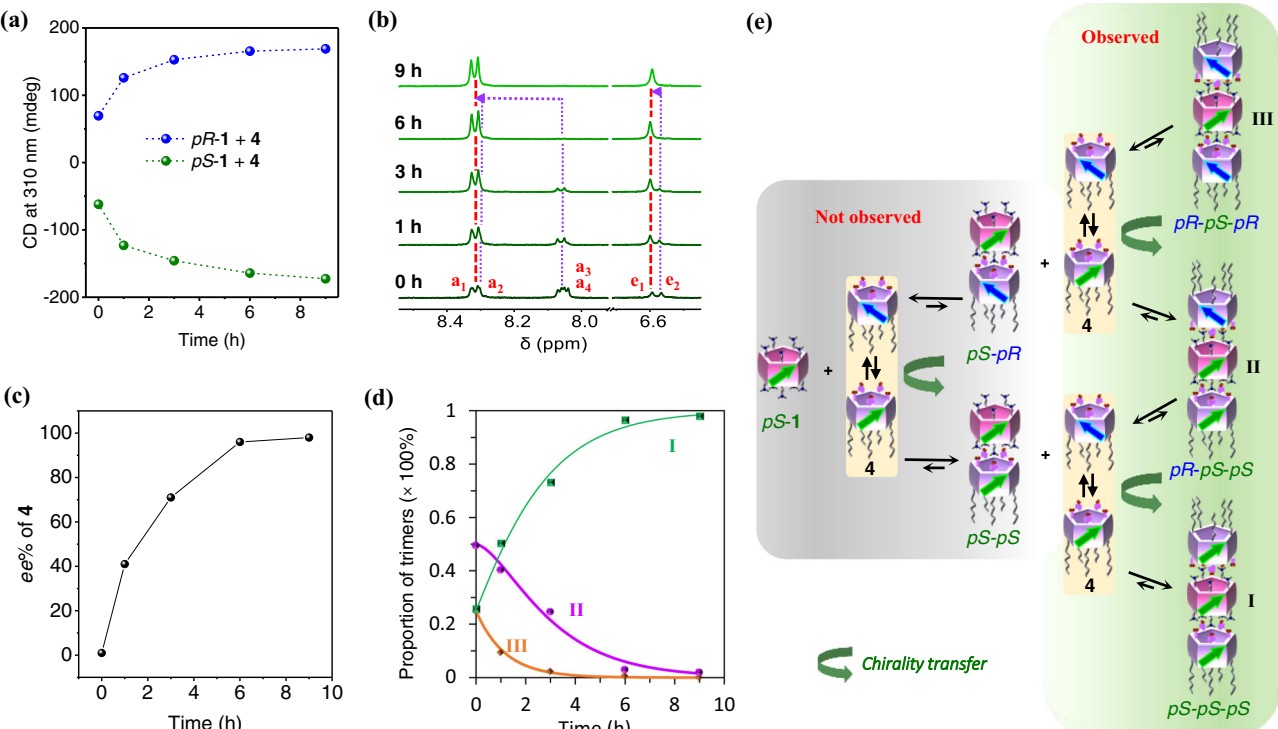

**Fig. 4 | Chirality transfer from enantiomeric 1 to 4 upon heating. a** CD intensities at 310 nm of mixtures of *pR*-1 or *pS*-1 with 4 upon heating at 50 °C. The concentrations of 1 and 4 were 0.05 and 0.1 mM, respectively. **b** Partial ¹H NMR spectra (400 MHz, CDCl₃) of a mixture of *pS*-1 and 4 upon heating at 50 °C. The concentrations of 1 and 4 were 0.5 and 1.0 mM, respectively. **c** Chiral induction of 4 and **d** the mole fractions of the three trimers upon heating at 50 °C. The *ee* of 4 was determined by the integrational ratio $(H_{a1} + H_{a2} - H_{a3} - H_{a4})/(H_{a1} + H_{a2} + H_{a3} + H_{a4})$. **e** Schematic of chirality transfer process from enantiomeric 1 to 4.

with a statistically random distribution and indicates that no immediate chirality transfer occurs in the mixture of *pS*-1 and 4. This is consistent with the low CD intensity observed in Fig. 3a, where an initial CD increase should be caused by the structural fixing of enantiomeric 1 by trimerization.

Chiral amplification is achieved upon heating the mixture of *pS*-1 and 4 at 50 °C. After ~9 h, the CD intensities of the mixtures of enantiomeric 1 and 4 reached their maxima (Fig. 4a and Supplementary Fig. 20), which are around 5 times higher than that of enantiomeric 1. Again, this most likely results from structural fixing via trimerization, which suppresses the swinging of pillar[5]arene units. The chiroptical responses of the systems also verified the above conclusion. At room temperature, the specific optical rotations of the 1:2 mixtures of *pS*-1 and *pR*-1 with 4 were $[\alpha]_D^{23} = -4.85 \pm 0.42$ and $+5.43 \pm 0.42$, respectively. After heating at 50 °C for 9 h, the optical rotations of the two systems changed to $-14.46 \pm 0.25$ and $+13.63 \pm 0.25$, respectively, almost three times as high as before heating. This suggested that heating led to the formation of homochiral assemblies.

The chirality transfer process was further monitored by variable temperature ¹H NMR analysis. Heterochiral trimers **II** and **III** gradually transform into homochiral trimer **I** upon heating (Fig. 4b and Supplementary Fig. 21). After around 9 h, the signals ascribed to the two heterochiral trimers are no longer observed and the *ee* of 4 exceeds 99% (Fig. 4c), indicating an eventually perfect chirality transfer in the mixture from *pS*-1 to 4. No species other than the three trimers are observed during the entire chirality transfer process. Therefore, we can represent the process simply as:

$$\text{Trimer III} \xrightarrow{k_1} \text{Trimer II} \xrightarrow{k_2} \text{Trimer I} \qquad (4)$$

From the change in the mole fractions of the three trimers shown in Fig. 4d, the reaction rates $k_{1,\,323K}$ and $k_{2,\,323K}$ in the above formula were determined to be $2.4 \times 10^{-4}\,s^{-1}$ and $1.3 \times 10^{-4}\,s^{-1}$, respectively. The

first transformation is faster than the second, because there are two *pR* form molecules of 4 in **III** and only one in **II**. Regrettably, it is difficult to determine the energy barriers for the two transformations because the rate of the chirality transfer process is difficult to precisely control as the temperature decreases (Supplementary Fig. 29). Details of the temperature effect are discussed in Supplementary Note 1.

A schematic of chirality transfer from *pS*-1 to 4 is shown in Fig. 4e. Typically, any heterochiral species in the mixture of *pS*-1 and 4 may dissociate at any mismatching position, and finally reassemble into a homochiral species through matching ion pairs after unit flipping of the dissociated *pR*-4, which constitutes an overall chirality transfer process. More specifically, when a *pR*-4 molecule in heterochiral trimer **III** dissociates from the assembly, the remaining heterochiral dimer can further dissociate into two monomeric pillar[5]arenes *pR*-4 and *pS*-1. Then, a homochiral dimer forms with the unit flipped *pR*-4 through matching ion pairs, which further assembles into **II** and **I**, completing a chirality transfer process. The chirality transfer from **II** goes through a similar process. However, these dimers could not be observed because of the strong ionic interactions between 1 and 4. The dissociated **III** immediately reassembles into **II** after unit flipping of *pR*-4, before assembling into **I** through a similar process.

With the length of the alkyl chains in between 2 and 4, pentaacid compound 3[49] completed the chirality transfer process in 10 min at 50 °C after mixing with *pS*-1 in a 1:2 molar ratio in chloroform (Supplementary Fig. 22), indicating that the chirality transfer slowed down under the same condition as increasing the length of the alkyl chains.

## Proposed mechanism

Overall, mixing of rotatable rim-differentiated acidic pillar[5]arenes 2, 3, and 4 with enantiomeric peraminopillar[5]arene 1 successfully generates homochiral trimeric nanotubes with high purity by chirality transfer during trimerization. The stabilization of trimers by the salt bridges in either homochiral or heterochiral trimers consumes other

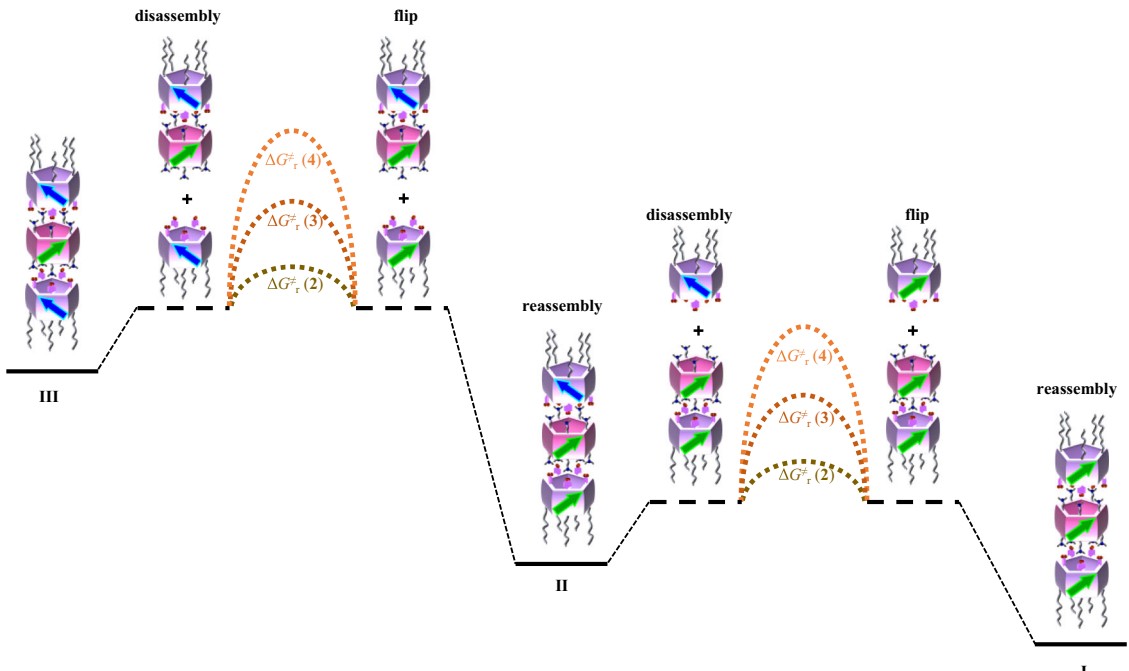

**Fig. 5 | Schematic representation of the effect of alkyl chain length on the kinetics of chirality transfer.** $\Delta G^{\ddagger}_r$ (**2**), $\Delta G^{\ddagger}_r$ (**3**), and $\Delta G^{\ddagger}_r$ (**4**) represent the rotation barrier of acidic pillar[5]arene **2**, **3**, and **4**, respectively. The resulting assemblies **III** (or **II**) from **2**, **3**, **4** require almost the same energy to dissociate one acidic pillar[5] arene, since the energy is mainly used against the same five pairs of ionic interaction. The chiral inversions of **2**, **3**, **4** contribute a major part of the conversion energy barrier of their assembled assemblies.

plausible species (i.e., dimers and monomers) in the system as soon as they are produced by trimer disassembly. Furthermore, the length of the alkyl chains on the rim-differentiated acidic pillar[5]arenes dramatically affects the speed of the chirality transfer process. We thus speculate that unit flipping of the acidic pillar[5]arenes (i.e., **2**, **3**, and **4**) after release from a trimer contributes to the energy barrier of transformation between trimers. This assumption that the association/dissociation processes are fast (compared with the unit flipping of pillar[5]arenes) is partly supported by the absence of heterochiral trimers when *rac*-**1** and **4** were mixed (Fig. 3c). The difference in the rotation abilities of **2**, **3**, and **4** (i.e., $\Delta G^{\ddagger}_r$ (**2**), $\Delta G^{\ddagger}_r$ (**3**) and $\Delta G^{\ddagger}_r$ (**4**) shown in Fig. 5) creates differences in the energy barriers between the trimers containing them, which results in the differences observed for mixtures of **2**, **3** or **4** with enantiomeric **1**. Using temperature variable $^1$H NMR, rotation barriers of **ester-2**, **ester-3**, and **ester-4**, which are unhydrolyzed ester-containing precursors, were determined to be 50.6, 56.2, and 59.4 kJ/mol, respectively (Supplementary Figs. 23–25). We and others have previously reported that the rotation of pillar[5] arenes is often completed by cooperative flipping of the five units[43,47,59,60], and this is also partly responsible for the fact that such a large difference in chirality transfer barrier can be realized by simply increasing the alkyl chain length from four carbon atoms to 12 carbon atoms. In addition, long alkyl chains may act as guests[61,62] in the cavity of the assemblies to stabilize them, thereby slowing their depolymerization and chirality transfer. We also investigated the solvent effect. In guest solvent dichloroethane, the chirality transfer rate was reduced, while in non-guest solvents, such as tetrachloroethane and tetrahydrofuran, the chirality transfer rates were as fast as in chloroform. Details of the length effect of alkyl chains and solvent effect are discussed in Supplementary Sections 6 and 7, respectively.

## Discussion

A combination of spectral data and computational results demonstrated chirality transfer from enantiomeric peraminopillar[5]arene **1** to the flippable acidic pillar[5]arenes **2**, **3**, and **4** via trimerization,

revealing the formation of discrete nanotubes with fivefold helices. Furthermore, the rate of chirality transfer can be successfully controlled by varying the length of the alkyl chains on the flippable achiral acidic pillar[5]arenes, and statistically random trimeric nanotubes form immediately upon mixing. Compared with most discrete chirality transfer systems based on coordination interactions and dynamic covalent bonds, which are strongly directional, this system has non-directional ionic interactions that help to maintain the random distribution of the trimers. The synergetic effects of the five strongly directional substituents (due to the planar chirality) on each rim of the non-flippable pillar[5]arene **1** ultimately allow real-time monitoring of the transformation from metastable heterochiral trimers to homochiral trimers. Although the precise mechanism of the chirality transfer process requires further investigation, it most probably involves the disassembly of mismatching assemblies in terms of planar chirality, unit flipping of the dissociated acidic pillar[5]arene, and reassembly between the planar chiral converted acidic pillar[5]arene and the chiral peraminopillar[5]arene **1**. Of these steps, the energy barrier to the flipping of the acidic pillar[5]arenes between the *pS* and *pR* forms seems to dominate the rate of chirality transfer. In this work, chiral tubular structures have been created through a chirality transfer strategy utilizing non-directional ionic interactions, and the intermediate heterochiral assemblies were captured. The discrete chirality transfer concept demonstrated in this work shows promise for the design of other dynamic chirality transfer systems, some of which may present clearer mechanisms and be easier to study, enriching our understanding of their properties even further.

## Methods

### Materials

All commercially available reagents and solvents were used as received. Compound **1** was synthesized by amination of a perbromopillar[5]arene with diethyl amine, and acidic pillar[5]arenes **2**, **3**, and **4** were synthesized by hydrolysis of their ester precursors obtained following a

general procedure to synthesized rim-differentiated pillar[5]arene. The synthetic details are shown in the Supplementary Information.

## Optical resolution of 1

Optical resolution of **1** was carried out on a JAI LaboACE LC-5060 HPLC apparatus equipped with a DAICEL CHIRALPAK® IE column ($\phi$ = 10 mm, $l$ = 250 mm). A sample of racemic **1** (5.0 mg) was dissolved in 5 mL of a mixture of $n$-hexane/ ethanol/ ethylenediamine (92.5/ 7.5/ 0.1, $v/v/v$). The solution was filtered, and 0.5 mL of the filtrate was injected. The separation was carried out using the same mixture of $n$-hexane/ethanol/ethylenediamine as eluent at a flow rate of 10 mL/min at 25 °C. The enantiomers were detected using a UV detector at 300 nm. For better separation, the sample was cycled three times on the column. The two fractions were collected by repeating the above operations. The solvent of each fraction was evaporated under reduced pressure at room temperature. The residue was redissolved in 5 mL chloroform, washed with DI water (5 mL × 3) to remove the residual ethylenediamine. (Note: Because of the presence of a large amount of ethylenediamine in the residue, the aqueous phase and the organic phase were very difficult to separate during the first washing. Overnight standing was necessary before the two could be completely separated). The organic phase was dried over anhydrous sodium sulfate after filtration and evaporated under reduced pressure at room temperature. Enantiomeric **1** was obtained as colorless viscous oil, which was kept at −10 °C prior to use, whereupon it solidified, becoming easy to weigh and use. Based on the positive and negative Cotton effect for the first and the second fractions observed in their CD spectra, the two fractions were assigned as $pR$-**1** and $pS$-**1**, respectively.

## Mass analysis

High-resolution ESI-MS was performed on a Thermo Fisher Scientific Exactive Plus mass spectrometer equipped with an UltiMate 3000 HPLC unit.

## NMR

$^{1}$H NMR spectra were recorded on JEOL JNM-ECS400, JNM-ECZ500R and JNM-ECA600P spectrometers. The DOSY analysis of a mixture of $pS$-**1** with **4** was performed on the JNM-ECA600P spectrometer at 25 °C. 2D COSY and 2D NOESY were also carried out to help to assign the signals of the mixture of $pS$-**1** with **4**. However, unexpected cross peaks resulted in complicated 2D NMR spectra, due to the chirality transfer at room temperature. At low temperatures (−20 °C), the spectra became broadened, and the cross-peaks were too weak to be analyzed.

## UV−Vis absorption

UV−Vis absorption spectra were recorded using a JASCO V-750 spectrophotometer equipped with a JASCO CTU-100 circulating thermostat unit to control the experimental temperature. All measurements were performed in chloroform at 25 °C. In all cases, the concentration of compound **1** was 0.05 mM, and 2 mm quartz cuvettes were used.

## CD

CD spectra were recorded on a JASCO J-1500 CD spectrometer equipped with a JASCO CTU-100 circulating thermostat unit to control the experimental temperature. In all cases, the concentration of compound **1** was 0.05 mM, and 2 mm quartz cuvettes were used.

## Optical rotations

Enantiomeric **1** (0.0005 mmol) and **4** (0.001 mmol) were mixed in chloroform (2.5 mL) to prepare the samples. Both the mixture of $pS$-**1** with **4** and that of $pR$-**1** with **4** were prepared. The optical rotations of the samples were measured on a Rudolph Research AUTOPOL IV Automatic Polarimete right after the preparation at room temperature and after heating the samples at 50 °C for 9 h. The values were averaged from 10 recorded data. The specific optical rotations were

determined by the equation: $[\alpha]^{23}_{D} = 100\ \alpha/c$, where $\alpha$ is the recorded optical rotation (in degree), and $c$ is the concentration of the samples (in mg/mL).

## Monitoring the trimer mixture

Time-dependent $^{1}$H NMR analyses of a mixture of $pS$-**1** with **4** at 50 °C were performed using the following method: 60 μL of a stock solution of $pS$-**1** in CDCl$_3$ (5 mM) and 120 μL of a stock solution of **4** in CDCl$_3$ (5 mM) were successively added to 420 μL CDCl$_3$. The mixture was transferred into an NMR tube and sealed. The spectrum of the mixture at 0 h was recorded as soon as possible at room temperature on the JEOL JNM-ECS400 spectrometer. The sample in the NMR tube was heated in an oil bath at 50 °C. The spectra were recorded again at 1, 3, 6, and 9 h. The temperature for recording spectra in all cases was 25 °C.

Time-dependent CD measurements of a mixture of $pS$-**1** with **2** (or **4**) at 50 °C were performed using the following method: The heater of the CD spectrometer was set to 50 °C. Then, 6 μL of a stock solution of $pS$-**1** in chloroform (5 mM) and 12 μL of a stock solution of **2** (or **4**) in chloroform (5 mM) were successively added in 582 μL of chloroform. The mixture was transferred to a quartz cuvette, which was sealed and immediately placed in the holder of the CD spectrometer. The spectrum of the mixture at 0 h was recorded as soon as possible. The sample was kept in the holder at 50 °C, and the spectra were recorded again at 1, 3, 6, and 9 h.

## Theoretical calculations

Theoretical calculations for structures of the homochiral and heterochiral trimers were carried out using ORCA 4.2.0 with the semi-empirical GFN2-xTB method and the SMD solvation model (chloroform).

## Nonlinear fitting of the chirality transfer process

Time-dependent changes of the mole fractions of trimers, [**I**], [**II**], and [**III**] were determined by $^{1}$H NMR spectroscopy. The mole fractions [**III**] and [**II**] were fitted to the equations of the two-step consecutive irreversible reaction kinetics with two parameters, $k_1$ and $k_2$, by nonlinear least-squares regression:

$$\textbf{Trimer III} \xrightarrow{k_1} \textbf{Trimer II} \xrightarrow{k_2} \textbf{Trimer I}$$

$$[\textbf{III}] = [\textbf{III}]_0 \exp(-k_1 t) \tag{5}$$

$$[\textbf{II}] = [\textbf{II}]_0 \exp(-k_2 t) + \frac{k_1[\textbf{III}]_0}{k_2 - k_1}\left(\exp(-k_1 t) - \exp(-k_2 t)\right) \tag{6}$$

$$[\textbf{I}] = 1 - [\textbf{III}] - [\textbf{II}] \tag{7}$$

where $[\textbf{III}]_0$ (= 0.25) and $[\textbf{II}]_0$ (= 0.5) are the mole fractions at $t$ = 0.

## Data availability

The data that support the findings of this study are available within the article and Supplementary Information files, and are also available from the corresponding author upon request. Supplementary Data 1 contains the optimized structures of the homochiral and heterochiral trimers in xyz format. Source data are provided with this paper.

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

## Acknowledgements

This work was supported by JSPS KAKENHI Grant Numbers JP15H00990, JP17H05148, JP18H04510, and JP20H04670 (Scientific Research on Innovative Areas, T.O.), JP19H00909 and JP22H00334 (Scientific Research (A), T.O.), JP21K14612 (Early Career Scientists, S.F.), JP20K22528 (Research Activity Startup, K.K.), JP21K20533 (Research Activity Startup, S.O.), JP22K14725 (Early Career Scientists, S.O.), JP21H01924 (Scientific Research (B), Y.N.), JST CREST Grant Number JPMJCR18R3 (T.O.), JST ERATO Grant Number JPMJER1903 (Y.N.), the Nanotechnology Platform Program of the Ministry of Education, Culture, Sports, Science and Technology (MEXT), and MEXT World Premier International Research Center Initiative (WPI), Japan. S.F. also thanks the financial support of Fundamental Research Funds for the Central Universities. The authors thank Dr. Masayuki Gon and Prof. Dr. Kazuo Tanaka (Kyoto University) for optical rotation measurements.

## Author contributions

Experimental data were taken by S.F., T.S., S.A., K.A., K.W., S.T., and N.O., and analysis and calculations were performed by S.F., T.S., K.K., S.O., Y.N., S.A., and T.O. All authors participated in discussion and writing of the manuscript.

## Competing interests

The authors declare no competing interests.
