## [Peer Review File · Nature Communications]

Real time chirality transfer monitoring from statistically random to discrete homochiral nanotubesREVIEWER COMMENTS

Reviewer #1 (Remarks to the Author):

In this manuscript contributed by Ogoshi et al., a series of homochiral pillar[5]arene-based trimeric assemblies was successfully built from an enantiopure deca-amine central unit and two penta-acid "end caps". By using a resolved optically-pure deca-amine, the authors showed that the chirality of the central unit can be effectively transferred to the two penta-acid units linked by salt bridges, resulting in homochiral trimer tubular structures. The chiral communications between these building blocks were studied in detail by NMR and circular dichroism spectroscopic techniques, as well as computational methods. Furthermore, by modifying the alkyl chains on the penta-acid to elevate the inversion barrier of the stereochemically-labile end caps, the authors demonstrated that the chirality transfer process can be slowed down enough, allowing for the real-time monitoring of the formation of the kinetically-trapped statistical diastereoisomeric mixtures and their transformations into the thermodynamically-stable homochiral trimer state over time.

For readers who are familiar with the Ogoshi group publications, the current manuscript can be seen as the last part of a "chiral nanotube" trilogy. The authors first demonstrated the construction of the salt bridged trimers as a racemic mixture in 2020 (*Angew. Chem. Int. Ed.* 2020, 59, 9309–9313; reference #47). Then the stereolabile penta-acid pillar[5]arene building block was regulated by chiral guests before assembly to obtain trimers with enantiomeric excess (*Chem. Sci.* 2021, 12, 3483–3488; not sure why this one was not included in the references section). This time, by resolving the deca-amine pillar[5]arene central unit by HPLC, enantiopure and homochiral trimers were finally obtained. While this manuscript appears to be a reasonable extension from the previous work as the authors gradually "perfected" the penta-acid and deca-amine salt bridge system, the technical difficulty is non-trivial. This is a great example for chiral sorting and chirality transfer of discrete molecular systems that deserves to be published.

With that being said, there are still some interesting points, especially regarding the mechanism, that the authors should offer further explanation.

An obvious question is that if the authors tried to measure the stereochemical inversion barriers of penta-acid compound 2 and 3? In reference #40 (*J. Org. Chem.* 2010, 75, 3268–3273), the inversion barriers of per-functionalized pillar[5]arenes were measured by DNMR. The delta G double dagger values for n-dodecyl and n-butyl substituted pillar[5]arenes are 63.2 and 51.8 kJ/mol (or 15.1 and 12.4 kcal/mol; in addition, the corresponding half-lives are roughly 10 msec and 0.2 msec), respectively. These might serve as reasonable estimations for the stereochemical inversion barriers of compound 3 and 2.

In the current manuscript the authors showed that whereas the pS 1 + 2 system generates immediately the homochiral trimer, it took 9 h at an elevated temperature (50 degree C) for the system of pS 1 + 3 to converge. Based on these observations, it seems that the barrier that the pS 1 + 3 system had to overcome is much larger than the difference between these delta G double dagger values. There should be some additional factors coming into play. The authors need to further augment both the data and discussions surrounding Fig 5 to make the mechanism more convincing.

In addition, here are some suggestions for the authors to further improve the manuscript:

Major points: In the introductory paragraph, the authors stated: "to the best of our knowledge, monitoring a chiral-transfer process from a statistically random assembly including heterochiral and homochiral units to a diastereomerically pure and thermodynamically stable homochiral assembly in a non-averaged discrete assembled system, i.e., a general and ideal chirality-transfer process, has never been reported." The literature examples the authors cited, including work published by Nitschke, Wu, and Sun, are mostly metal-coordinated systems that are highly labile. Therefore the kinetically-trapped diastereomeric mixtures are short-lived and possibly could not be observed. On the other hand, the self-assembly processes of dynamic covalent systems, such as imine cages, are

highly reversible, but not as labile as the corresponding systems. In many publications by Michael Mastalerz and Xiaoyu Cao, “errors” happening in the assembly process can be identified or even isolated, and then the overall system can converge into the most thermodynamically-stable product after being “annealing” at higher temperatures. For particular example, please see the assembly of face-rotating polyhedron FRP-2 (Cao et al. Nat. Commun. 2016, 7, 12469). It would be safer for the authors not to claim “never been reported” in the manuscript.

Minor points:

Since this article deals exclusively with pillar[5]arene derivatives, it would be more convenient for the readers if “pillar[5]arenes” are used throughout. Most “pillar[n]arenes” in the manuscript can be changed to “pillar[5]arenes.”

In literature, there is no clear boundary in terms of energy barrier between conformers and atropisomers. Since the authors can already effectively achieve chiral resolution of the two enantiomers of the deca-amine, it would be more appropriate to use “atropisomers” instead of “isolable conformers” in the text.

Reviewer #2 (Remarks to the Author):

This manuscript by Ogoshi et al. described a new approach to the construction of homechiral nanotubes using supramolecular assembly of the planar chiral pillar[5]arenes with ionic interactions. The authors experimentally and computationally explored an interesting sequence of mismatching assembly, dissociation, unit flapping and reassembly to disclose the real-time chirality transfer from statistically random hetrotrimers to diastereomerically pure assembly of homotrimers. All the data were clearly discussed and I like the writing of this manuscript. Compounds including the possible intermediates were well characterized by a number of spectroscopic studies.

This work is important because the discrete chirality-transfer concept and the novel mechanism represent an advance in chiral supramolecular nanotubes that will in principle be of broad interest to the readership of pillararene chemistry. Therefore, it is the opinion of this reviewer that the manuscript would be suitable for publication in Nature Communications after addressing some issues that I feel would improve it:

1. The author claim that enantiomeric 1 is stable at room temperature and the date fitting determined by Eyring equation was discussed in more detail in SI. The readability of this manuscript could be further enhanced by including brief comments on the resulting thermodynamic parameters in the main text.

(2) The thermal stability of pS-1 was only examined by heating solutions up to 55 °C which seems very low (slightly higher than ambient conditions) and not really sufficient for related applications. Despite an expectation that the “swing effect” might be somewhat intensified, the investigations at even higher T such as 100 °C are essentially important, which could be nicely achieved by HPLC techniques.

(3) The chirality transfer process from enantiomeric 1 to 3 is complete in 48 h which is a long period of time. In my opinion, rate of this process should be considerably enhanced by truncating length of the alkyl chains substituted on 3. Length tuning with number of carbons between C4 and C12 would allow studies of the energy barriers for the conversion from trimer III to II and for that from II to I, which is an absence in the present work.

(4) Chiroptical responses to the chirality transfer processes should also be reflected in optical rotations, which should appear to be information in support of the CD spectra.

(5) Out of my further curiosity, do these tubular homotrimers of pillar[5]arenes show possibility to bind guest species? Especially for the selective complexation of chiral molecules?

(6) The high-resolution ESI-mass data for two homotrimers require some attention. The peaks assigned to species of higher molecular weight are difficult to distinguish from noise signals. This reviewer suggests blow up those peaks together with the full spectra in Supplementary Fig. 13 and Fig. 15.

(7) Most of the chiral transfer experiments were monitored in chloroform, is there any evidence for solvent dependence of supramolecular assembly?

Reviewer #3 (Remarks to the Author):

Fa et al. reports on the formation of trimeric nanotubular assemblies formed by ionic interactions between the amine-based rims of a pillar[5]arene and the acidic functionalities of a similarly modified pillar[5]arene. The most remarkable feature is that by using a chiral peraminopillar[5]arene the authors were able to monitor in real-time from a statistically random to a diastereomerically pure assembly. My enthusiasm for the publication of this work in its current form in Nature Communications got diminished when learning about the author's prior work, ref 47, where the formation of trimeric assemblies using the same building blocks was demonstrated. Thus, my view of this work is incremental in nature and lacks the novelty and urgency suitable for a manuscript in this journal. Overall, the work related to chirality transfer is elegantly done and presented, however it may be better suited in a more technical journal.

Reviewer #1 (Remarks to the Author):

Comment 1: For readers who are familiar with the Ogoshi group publications, the current manuscript can be seen as the last part of a “chiral nanotube” trilogy. The authors first demonstrated the construction of the salt bridged trimers as a racemic mixture in 2020 (Angew. Chem. Int. Ed. 2020, 59, 9309–9313; reference #47). Then the stereolabile penta-acid pillar[5]arene building block was regulated by chiral guests before assembly to obtain trimers with enantiomeric excess (Chem. Sci. 2021, 12, 3483–3488; not sure why this one was not included in the references section). This time, by resolving the deca-amine pillar[5]arene central unit by HPLC, enantiopure and homochiral trimers were finally obtained. While this manuscript appears to be a reasonable extension from the previous work as the authors gradually “perfected” the penta-acid and deca-amine salt bridge system, the technical difficulty is non-trivial. This is a great example for chiral sorting and chirality transfer of discrete molecular systems that deserves to be published.

Response: We sincerely thank the reviewer for a complete summary, support and high evaluation of our work. We also appreciate the comment of the reviewer to improve this paper. The reviewer expressed confusion as to why we did not cite the work of “Chem. Sci. 2021, 12, 3483–3488” in our citations. Our original intention was because the above work involved the regulation between diastereomers, while the focus of this submitted manuscript is on the transformation between pure enantiomers. In order not to confuse readers, we did not cite this article in the previous submitted manuscript. However, as reminded by the reviewer, we recognize the important transitional role of the above work in our entire series of work, and therefore, in the new revised version, we have included the explanation and citation of this part of the work, with the citation 50 appears in the introduction section, which has been highlighted in yellow text as follows:

“Subsequently, we tried to use acidic pillar[5]arenes containing five stereogenic carbons to achieve the construction of chiral nanotubes by regulating the diastereomeric excess of the acidic pillar[5]arenes using guest molecules, but the structural details of the chiral assemblies were still not clear.⁵⁰”

Comment 2: An obvious question is that if the authors tried to measure the stereochemical inversion barriers of penta-acid compound 2 and 3? In reference #40 (J. Org. Chem. 2010, 75, 3268–3273), the inversion barriers of per-functionalized pillar[5]arenes were measured by DNMR. The ΔG^\ddagger values for n-dodecyl and n-butyl substituted pillar[5]arenes are 63.2 and 51.8 kJ/mol (or 15.1 and 12.4 kcal/mol; in addition, the corresponding half-lives are roughly 10 msec and 0.2 msec), respectively. These might serve as reasonable estimations for the stereochemical inversion barriers of compound 3 and 2.

Response: We are grateful to the reviewer for this very professional comment to improve this paper. Determination of the inversion barriers of penta-acid compounds is of great help in understanding the mechanism of the difference in chirality transfer process.

In order to make the conclusion more credible, we synthesized a new penta-acid compound while revising this paper, one rim of which is the same as **2** and **3** with 5 benzoic acid groups, and the other rim is 5 octyl groups. To make the text more fluent, we have changed the numbering of the compounds: The penta-acid compound with 5 butyl groups is still named **2**, the penta-acid compound with 5 octyl groups is named **3**, and the penta-acid compound with 5 dodecyl groups is changed to **4** (, whose number was **3** in the previous version).

Unfortunately, however, we cannot directly measure the inversion barriers of compounds **2**, **3** and **4** in chloroform. This is because the molecules contain carboxyl groups, which can act as both hydrogen bond acceptors and hydrogen bond donors, and the molecules will assemble in pairs through hydrogen bonds, which has been observed in our previous work (ref 48 and 49).

To determine the inversion barriers suggested by the reviewer, we selected the analogs of **2**, **3** and **4**, that is, their unhydrolyzed ester-containing precursors as subjects, namely, **ester-2**, **ester-3**, and **ester-4**, respectively. The volumes of the upper and lower rims of these analogs are similar to **2**, **3** and **4**, respectively, thus can represent **2**, **3** and **4**. The ^1H NMR signal of the methylene of benzyl groups in molecules **ester-2**, **ester-3**, and **ester-4** can split at lower temperatures and coalesce at higher temperatures due to the molecular inversion rate. Therefore, with the help of temperature-variable ^1H NMR technology, we can determine their inversion barriers. The inversion barriers of **ester-2**, **ester-3** and **ester-4** are determined to be 50.6, 56.2 and 59.4 kJ/mol, respectively, which are slightly lower than the corresponding pillar[5]arenes whose upper and lower rims are both alkyl groups of the same length (J. Org. Chem. 2010, 75, 3268–3273).

We believe that the five bulky benzoate groups at one rim of the molecules may make the opening sizes of the two rims of the pillar[5]arenes **ester-2**, **ester-3** and **ester-4** different. The molecule is folded towards the rim of alkyl chains, thus making it easier for the molecule to flip through the alkyl side.

The related content is mentioned in the text as follows and discussed in detail in the fourth part of SI:

“Using temperature-variable ^1H NMR, rotation barriers of **ester-2**, **ester-3** and **ester-4**, which are unhydrolyzed ester-containing precursors, were determined to be 50.6, 56.2 and 59.4 kJ/mol, respectively (Supplementary Figs. 23, 24 and 25).”

Comment 3: In the current manuscript the authors showed that whereas the pS 1 + 2 system generates immediately the homochiral trimer, it took 9 h at an elevated temperature (50 degree C) for the system of pS 1 + 3 to converge. Based on these observations, it seems that the barrier that the pS 1 + 3 system had to overcome is much larger than the difference between these delta G double dagger values. There should be some additional factors coming into play. The authors need to further augment both the data and discussions surrounding Fig 5 to make the mechanism more convincing.

Response: We are grateful to the reviewer for this very professional comment to improve this paper. The chirality-transfer rate of the mixture of pS-1 and 3 is between that of 2 and 4, which is as expected. However, given the single-molecule inversion barriers of 2, 3 and 4 are not significantly different, we still cannot fully understand all the influencing factors in the chirality transfer process. We proposed some possibilities on these observations as following:

1. The rotation barriers of their unhydrolyzed ester precursors of compounds 2, 3 and 4 do not really reflect their own rotation barriers. The molecules of 2, 3 and 4 in chloroform could interact with each other through five pairs of hydrogen bonds, producing dimers. As the length of alkyl chains on one rim of the pillar[5]arene increasing, the steric hindrance of that rim increases consequently. This may cause the five benzoic acid groups on the other rim more compact, so that the strength of its intermolecular or intramolecular hydrogen bond network will be enhanced, and then the rotation barrier of the acidic pillar[5]arene will be raised.

2. Due to the large steric hindrance of the five benzoic acid groups, the acidic pillar[5]arenes 2, 3 and 4 may not perfectly pillar-shaped, so the longer the alkyl chain on the opposite rim may make the entire pillar[5]arene molecule closer to the pillar-shape, thereby adding additional stability to the trimeric nanotubes formed after mixing with enantiomeric 1, making the assemblies less prone to dissociation.

3. Although the alkyl chains are not the ideal guests of the cavity of pillar[5]arenes, they can still be recognized by pillar[5]arenes to some extent. Thus, longer alkyl chains can prevent the rotation of the acidic pillar[5]arenes by either entering its own cavity or inserting into the cavity of other pillar[5]arenes, and even make the acidic pillar[5]arenes less likely to dissociate from their trimeric nanotubular assemblies.

Our attempt to investigate the mechanism in depth is presented in the fourth part of the SI, titled "Mechanism of the chirality-transfer process". There, we first discussed the effect of the

alkyl chain length on the inversion of the single molecule of **2**, **3** and **4**, and then we discussed the mechanism in detail.

Comment 4: In the introductory paragraph, the authors stated: “to the best of our knowledge, monitoring a chiral-transfer process from a statistically random assembly including heterochiral and homochiral units to a diastereomerically pure and thermodynamically stable homochiral assembly in a non-averaged discrete assembled system, i.e., a general and ideal chirality-transfer process, has never been reported.” The literature examples the authors cited, including work published by Nitschke, Wu, and Sun, are mostly metal-coordinated systems that are highly labile. Therefore the kinetically-trapped diastereomeric mixtures are short-lived and possibly could not be observed. On the other hand, the self-assembly processes of dynamic covalent systems, such as imine cages, are highly reversible, but not as labile as the corresponding systems. In many publications by Michael Mastalerz and Xiaoyu Cao, “errors” happening in the assembly process can be identified or even isolated, and then the overall system can converge into the most thermodynamically-stable product after being “annealing” at higher temperatures. For particular example, please see the assembly of face-rotating polyhedron FRP-2 (Cao et al. Nat. Commun. 2016, 7, 12469). It would be safer for the authors not to claim “never been reported” in the manuscript.

Response: We thank this reviewer very much for pointing out this problem. The reviewer's suggestion is very constructive. In fact, we were previously aware of some of these references mentioned by the reviewer. As the reviewer put it, "In many publications by Michael Mastalerz and Xiaoyu Cao, 'errors' happening in the assembly process can be identified or even isolated, and then the overall system can converge into the most thermodynamically-stable product after being " annealing” at higher temperatures.” In these cases, homochiral assemblies and their heterochiral isomers were assembled from achiral ligands, mostly not involving the concept of chiral transfer, which is slightly different from the focus of the current work. However, we recognize that the expression "never been reported" is not rigorous. Therefore, we have changed the above expression to "remains challenging". At the same time, we also cite the paper mentioned by the reviewer as ref. 35 for comparison with our work in the main text as follows: “On the other hand, both homochiral assemblies and their heterochiral isomers can be observed at the beginning of mixing in several self-assembled systems, but not by a chirality-transfer strategy.³⁵”

Comment 5: Since this article deals exclusively with pillar[5]arene derivatives, it would be more convenient for the readers if “pillar[5]arenes” are used throughout. Most “pillar[n]arenes” in the manuscript can be changed to “pillar[5]arenes.”

Response: We greatly appreciate this valuable comment and we have changed all “pillar[n]arenes” in the manuscript to “pillar[5]arenes”.

Comment 6: In literature, there is no clear boundary in terms of energy barrier between conformers and atropisomers. Since the authors can already effectively achieve chiral resolution of the two enantiomers of the deca-amine, it would be more appropriate to use “atropisomers” instead of “isolable conformers” in the text.

Response: We greatly appreciate this valuable comment and fully agree with it. Accordingly, we have changed the description “isolable conformers” in the manuscript to “atropisomers”.

Reviewer #2 (Remarks to the Author):

We highly appreciate the comments of the reviewer to improve this paper. According to the comments of the reviewer, we revised our manuscript as follows:

Comment 1: The author claim that enantiomeric **1** is stable at room temperature and the date fitting determined by Eyring equation was discussed in more detail in SI. The readability of this manuscript could be further enhanced by including brief comments on the resulting thermodynamic parameters in the main text.

Response: We thank this reviewer very much for pointing out this problem. Stability is a very important property of enantiomeric **1**. Therefore, according to the reviewer's opinion, we have discussed the stability parameters of **1** in the main text. The Gibbs energy of activation at 25 °C ($\Delta G^\ddagger_{25^\circ\text{C}}$), shown in the SI by Eyring equation was also included in the main text. And together with answering the reviewer's next question, the stability of **1** is discussed more abundantly. Relevant content is highlighted with yellow in the main text as follows:

“The enantiomeric **1** is stable at room temperature as the Gibbs energy of activation at 25 °C ($\Delta G^\ddagger_{25^\circ\text{C}}$) was determined to be 101 kJ/mol by Eyring plot at 40–55 °C. Moreover, the CD intensity of enantiomeric **1** does not significantly decrease over four weeks at room temperature (see Supplementary Section 1.2 for details).”

Comment 2: The thermal stability of pS-1 was only examined by heating solutions up to 55 °C which seems very low (slightly higher than ambient conditions) and not really sufficient for related applications. Despite an expectation that the “swing effect” might be somewhat intensified, the investigations at even higher T such as 100 °C are essentially important, which could be nicely achieved by HPLC techniques.

Response: We thank this reviewer very much for this valuable suggestion. According to this suggestion, we examined the stability of *pS-1* at 100 °C. Taking into account the factors of solubility and boiling point, we used tetrachloroethane as the solvent. Unfortunately, too high a temperature (such as 100 °C) was found to rapidly racemize *pS-1*. The CD signal of *pS-1* at 100 °C completely disappeared within 3 hours, and no signal was observed even when it returned to room temperature. As the reviewer point out, this limitation may reduce the practical application of the system. In future work, we can increase the stability of **1** by increasing the steric hindrance of the substituents on the upper and lower rims of **1**. We have already started related systematic work.

Regarding this part, we have made changes in the main text and Supplementary Section 1.2, and added Fig 9 in the SI, so the number of pictures in SI after that has been changed. Relevant content is highlighted in yellow in the main text as follows:

“However, too high a temperature will rapidly racemize enantiomeric **1**, as the CD signal of enantiomeric **1** at 100 °C in tetrachloroethane completely disappeared within 3 hours, and no signal was observed even when it cooled back to room temperature (Supplementary Fig. 9).”

Comment 3: The chirality transfer process from enantiomeric **1** to **3** is complete in 48 h which is a long period of time. In my opinion, rate of this process should be considerably enhanced by truncating length of the alkyl chains substituted on **3**. Length tuning with number of carbons between C4 and C12 would allow studies of the energy barriers for the conversion from trimer III to II and for that from II to I, which is an absence in the present work.

Response: We thank this reviewer very much for this valuable suggestion. Following the reviewer's comment, we synthesized acidic pillar[5]arene with 8 carbon atoms at the alkyl rim, named it **3**, measured its inversion energy barrier, and also studied the chirality-transfer behavior of the system after it was mixed with *pS-1*. And compound **3** in the previous version with 5 dodecyl groups are changed to number **4** in the revised version consequently. The results show that the chirality-transfer rate of the mixture of *pS-1* and **3** is between that of **2** and **4**, which is reasonable. We have made relevant changes in the main text as follows and detailedly explained it in the Supplementary Information:

“With the length of the alkyl chains in between **2** and **4**, penta-acid compound **3**⁴⁹ completed the chirality-transfer process in 10 min at 50 °C after mixing with *pS-1* in a 1:2 molar ratio in chloroform (Supplementary Fig. 22), indicating that the chirality-transfer slowed down under the same condition as increasing the length of the alkyl chains.”

Comment 4: Chiroptical responses to the chirality transfer processes should also be reflected in optical rotations, which should appear to be information in support of the CD spectra.

Response: We thank this reviewer very much for this valuable suggestion and fully agree with the reviewer's comment. Following the comment, we have measured the optical rotation change of the

mixture of enantiomeric **1** and **4** in a 1:2 molar ratio in chloroform. system. We found that the optical rotation of the system after heating at 50 °C is roughly three times that of room temperature, which is in good agreement with the results of our previous CD and NMR experiments. Relevant texts has been added to the “Results” part and the “Methods” part as follows:

“The chiroptical responses of the systems also verified the above conclusion. At room temperature, the specific optical rotations of the 1:2 mixtures of *pS*-**1** and *pR*-**1** with **4** were $[\alpha]^{23}_D = -4.85 \pm 0.42$ and $+5.43 \pm 0.42$, respectively. After heating at 50 °C for 9 hours, the optical rotations of the two systems changed to -14.46 ± 0.25 and $+13.63 \pm 0.25$, respectively, almost three times as high as before heating. This suggested that heating led to the formation of homochiral assemblies.”

“**Optical rotations.** Enantiomeric **1** (0.0005 mmol) and **4** (0.001 mmol) were mixed in chloroform (2.5 mL) to prepare the samples. Both the mixture of *pS*-**1** with **4** and that of *pR*-**1** with **4** were prepared. The optical rotations of the samples were measured on a Rudolph Research AUTOPOL IV Automatic Polarimeter right after the preparation at room temperature and after heating the samples at 50 °C for 9 h. The values were averaged from 10 recorded data. The specific optical rotations were determined by the equation: $[\alpha]^{23}_D = 100 \alpha/c$, where α is the recorded optical rotation (in degree), and c is the concentration of the samples (in mg/mL).”

Comment 5: Out of my further curiosity, do these tubular homotrimers of pillar[5]arenes show possibility to bind guest species? Especially for the selective complexation of chiral molecules?

Response: We are very grateful to the reviewer for this constructive question. In our previous report, such trimeric nanotubular assemblies bind dibromobutane stepwise, a common guest molecule of the cavity of pillar[5]arenes. But according to our experience, the cavity of pillar[5]arenes only show strong binding ability to linear molecules because of the cavity size, and the binding of chiral molecules is very weak. We and other researchers attempted to achieve selective recognition of chiral guests through pillar[6]arenes with larger cavities. Fabrication of pillar[6]arene-based trimeric nanotubes are ongoing in our lab and related work to this question will be carried out in the future.

Comment 6: The high-resolution ESI-mass data for two homotrimers require some attention. The peaks assigned to species of higher molecular weight are difficult to distinguish from noise signals. This reviewer suggests blow up those peaks together with the full spectra in Supplementary Fig. 13 and Fig. 15.

Response: We thank the reviewer very much for this suggestion. The reviewer suggested us to blow up the mass spectra in Supplementary Fig. 13 and Fig. 15. we took the liberty to assume that the reviewer was referring to the mass spectra in Supplementary Fig. 10 and Fig. 15, and we have made changes following the reviewer's suggestion. Since we added other figures before these two figures in the SI, the above two figures are marked as Fig. 11 and Fig. 16 respectively in the modified SI.

Comment 7: Most of the chiral transfer experiments were monitored in chloroform, is there any evidence for solvent dependence of supramolecular assembly?

Response: We are very grateful to the reviewer for this constructive question. The solvent effect should have an important influence on the self-assembly of the system. Unfortunately, due to solubility and assembly issues, we do not have many solvents of choice for this system. For example, large polar solvents, such as DMSO, methanol, etc., will greatly weaken the assembly ability between amino- and acidic-pillar[5]arenes, and are not conducive to the dissolution of alkyl chains in acidic-pillar[5]arenes. Therefore, halogenated solvents are a better choice for this system.

We chose dichloroethane (DCE) and tetrachloroethane (TCE) as solvents. In order to increase the solubility, we added 1% chloroform to each of the above solvents. In addition, the system has good solubility in tetrahydrofuran (THF), so we also chose THF as the solvent. In summary, we investigated three more solvent systems, namely DCE + 1% CHCl₃, TCE + 1% CHCl₃ and THF.

We selected the 1:2 mixture of *pS*-**1** and **2** as the representative system. This is mainly due to the fast chirality transfer of the mixture of *pS*-**1** and **2** in chloroform. Changing the solvent for this system may slow down the chirality transfer process described above. We found that the chirality transfer was completed after 1 h in the solvent mixture of DCE + 1% CHCl₃, while it finished immediately after the mixing in the other two solvent systems. It probable that DCE molecules can enter the cavity of pillar[5]arene **2** as guests, thereby reducing the rate at which it is induced by the chirality of **1**. However, THF and TCE are bulky like chloroform and cannot serve as guest molecules of pillar[5]arenes, so the chirality transfer in them is as fast as in chloroform. Thus, our present few examples show that the solvent effect mainly comes from whether the solvent molecule can enter the cavity of pillar[5]arenes as a guest.

The related content is discussed in detail as the fifth part of SI and briefly in the main text as follows: “We also investigated the solvent effect. In guest solvent dichloroethane, the chirality-transfer rate was reduced, while in non-guest solvents, such as tetrachloroethane and tetrahydrofuran, the chirality-transfer rates were as fast as in chloroform. Details of the length effect of alkyl chains and solvent effect are discussed in Supplementary Sections 5 and 6, respectively.”

Reviewer #3 (Remarks to the Author):

Comment: Fa et al. reports on the formation of trimeric nanotubular assemblies formed by ionic interactions between the amine-based rims of a pillar[5]arene and the acidic functionalities of a similarly modified pillar[5]arene. The most remarkable feature is that by using a chiral peraminopillar[5]arene the authors were able to monitor in real-time from a statistically random to a diastereomerically pure assembly. My enthusiasm for the publication of this work in its current form in Nature Communications got diminished when learning about the author’s prior work, ref 47, where

the formation of trimeric assemblies using the same building blocks was demonstrated. Thus, my view of this work is incremental in nature and lacks the novelty and urgency suitable for a manuscript in this journal. Overall, the work related to chirality transfer is elegantly done and presented, however it may be better suited in a more technical journal.

Response: First of all, we are extremely grateful to the reviewer for his or her affirmation of the technical aspects of this work. The reviewer pointed out that this work is based on our previous work ref 47 (now ref 48) and thus lacks novelty. If considered from a molecular design perspective, we partially agree with this reviewer. But at the same time, as we pointed out in the introduction part of the article, the successful monitoring of the assembly process of chiral assembly systems is of great significance to gain a deeper understanding of the mechanism of chirality. This work is based on this important scientific question. We used the molecular design in ref 48 as a platform and carefully designed a series of experiments related to chirality transfer to test something completely different from that reported in ref 48. Thanks to this molecular platform, we have observed chiral transfer processes that were previously difficult to monitor throughout the process. Therefore, we take the liberty to consider the research work of this paper as novel.

In addition to making point-by-point revisions to the comments raised by the reviewers, we also made revisions to several minor errors that we found by ourselves.

REVIEWERS' COMMENTS

Reviewer #1 (Remarks to the Author):

In the revised manuscript, in addition to the compounds 2 and 4 originally included, the authors have added new penta-acid rim-differentiated pillar[5]arene with n-octyl chains (compound 3) and conducted dynamic NMR studies to determine the inversion barriers of the ester derivatives of all three. As expected, these compounds have similar inversion barriers ranging between 50–60 kJ/mol. The huge difference in the chirality transfer rates (0 sec vs. 9 h) in their corresponding different trimeric assemblies still remains a mystery. This interesting phenomenon is definitely worth further investigations, but this would not affect the integrity of this current manuscript. There are also some new additions of other experimental results and corrections, which help to enhance the consistency and quality of this work. The authors have already properly addressed all questions and suggestions in my previous report.

Reviewer #2 (Remarks to the Author):

The authors nicely addressed this reviewer's comments, and the acceptance of this manuscript is recommended for publication.